# Detection of Multi-Modal Doppler Spectra. Part 1: Establishing Characteristic Signals in Radar Moment Data

Sarah Wugofski<sup>1</sup>, Matthew R. Kumjian1<sup>1</sup>, Mariko Oue<sup>2</sup>, and Pavlos Kollias<sup>2,3</sup>

Correspondence: Sarah Wugofski (sjw5417@psu.edu)

Abstract. Vertically pointing millimeter-wavelength radars provide a wealth of information about cloud and precipitation particle properties. Doppler spectral data can inform on how particles of varying vertical velocities contribute to total backscattered power observed. It is more computationally cost effective to process moment data instead of spectra data, but doing so leaves valuable information on the cutting room floor. To confidently identify a multi-modal spectra event, in which two or more modes are present within a layer, Doppler spectral data are essential. This means long-term identification of layers featuring multi-modal spectra can be cost prohibitive. To address this, we explore three multi-modal spectra cases from winter precipitation events to determine characteristic signatures of these layers in the moment data averaged over short time periods (~145 s) and explore how these layers differ from the rest of the vertical profiles. We find that the mean spectrum width and the standard deviation of mean Doppler velocity can be used to determine whether or not a layer is multi-modal. In particular, multi-modal layers in mixed-phase and ice clouds feature larger mean spectrum width (exceeding 0.17 m s<sup>-1</sup>) and smaller standard deviation of the mean Doppler velocity (below 0.1 m s<sup>-1</sup>). In Part 1 of this study, the identification criteria and methods are described. In Part 2, we perform a verification of the method for three years of vertically pointing radar data, and explore the meteorological conditions associated with identified multi-modal spectral events.

## 15 1 Introduction and Background

Radar sampling volumes typically contain millions of hydrometeors, each of which may move according to the local wind speed, turbulence, updrafts/downdrafts, etc. Hydrometeor motion, when projected along the radar wave's propagation direction, may vary, leading to a dispersion of radial velocities within the sampling volume. Further, waves backscattered from each hydrometeor in the sampling volume interfere, resulting in a combined received signal whose amplitude and phase may fluctuate from pulse to pulse. Because the hydrometeors' locations and sizes within the radar sampling volume (which spans many wavelengths in range) are random, these received signals can be considered random signals (Doviak and Zrnić, 1993). Assuming ergodicity, statistical properties of the sampled hydrometeors can

<sup>&</sup>lt;sup>1</sup>Department of Meteorology & Atmospheric Science, The Pennsylvania State University, University Park, Pennsylvania, USA

<sup>&</sup>lt;sup>2</sup>School of Marine & Atmospheric Sciences, Stony Brook University, State University of New York, Stony Brook, New York, USA

<sup>&</sup>lt;sup>3</sup>Environmental & Climate Sciences Division, Brookhaven National Laboratory, Upton, New York, USA

be obtained from sufficient time averages (i.e., averaging multiple pulses). The frequency distribution of the random signals may be obtained by taking the Fourier transform of the signal's autocorrelation function; this frequency distribution is known as the power spectrum. When converted from frequency to radial velocity, one obtains the Doppler spectrum: the power- (or reflectivity-) weighted distribution of radial velocities within in sampling volume (Doviak and Zrnić, 1993).

The Doppler spectrum shows the contribution to the overall received signal power from hydrometeors as a function of their radial velocity. The examples in Fig. 1 are spectrograms (i.e., graphical representations of the Doppler spectrum) taken from the U.S. Department of Energy (DOE) Atmospheric Radiation Measurement (ARM) program's Ka-band ARM Zenith-pointing Radar (KAZR) at the Southern Great Plains (SGP) site. Herein, we make use of the convention that negative velocities are towards the radar, meaning hydrometeors are descending for these vertically pointing radars. The black curve shows an approximately Gaussian-shaped peak in co-polar spectral power centered on velocities between 0 and  $-2 \text{ m s}^{-1}$ ; the radar's noise floor is at approximately  $-40 \text{ dBm (m s}^{-1})^{-1}$  in this example. From such a Doppler spectrum, one can visualize how particles of differing velocities contribute to the total backscattered power observed. Two radar moments obtained by integration over the Doppler spectrum are mean Doppler velocity (MDV) and spectrum width (SW). The bi-modal spectrogram (shown in green) is broader than the one with a single mode; this translates to a larger SW. When a secondary peak is offset to slower velocities (as shown in Fig. 1), it may noticeably affect the MDV by shifting it towards smaller velocities. The magnitude of this shift depends on the total power of the secondary mode relative to the rest of the spectrum. Thus, in situations featuring bi-modal spectra, it can be expected that these spectral signals may affect those radar moment variables, as well.

Although hydrometeors in a radar sample volume may exhibit different Doppler velocities when viewed at low elevation angles, particularly in highly sheared environments (e.g., Wang et al., 2019; Hernandez and Chandrasekar, 2023), most often Doppler spectral analysis is used for high antenna elevation angles, including  $45^{\circ}$  (e.g., Moisseev et al., 2004; Spek et al., 2008; Mak and Unal, 2024) and  $90^{\circ}$  (i.e., vertically pointing radars; e.g., Kollias et al., 2002; Moisseev et al., 2006; Li and Moisseev, 2019; Kumjian et al., 2020). For vertically pointing radars, the measured Doppler velocities are closely related to the hydrometeor fall speeds. Different sizes and types of precipitation tend to have different fall speeds (e.g., Lamb and Verlinde, 2011). Larger and denser hydrometeors, such as hail, will fall very fast (e.g., up to 50-60 m s<sup>-1</sup>; Heymsfield et al., 2018), whereas smaller hydrometeors, such as pristine ice crystals, will fall much slower. In the snowstorm example shown in Fig. 1, the fall speeds were all  $

Figure 1. Example of a co-polar power spectrum from the KAZR at the ARM Southern Great Plains (SGP) site near Lamont, Oklahoma, taken on 13 December 2020 at 1251 UTC during a snow event. The black curve represents a spectrogram taken at 3 km above radar level; the green at 2.2 km above radar level. Note that negative velocities correspond to descending hydrometeors.

of drizzle, riming, and secondary ice generation using spectral reflectivity, spectral linear depolarization ratio, and MDV. In particular, multi-modal radar spectra indicate that multiple particle types and/or sizes with distinct fall speeds are present, which can be particularly informative when trying to deduce active microphysical processes (e.g., Kalesse et al., 2016; Billault-Roux et al., 2023).

Because signals from clouds and precipitation can be distinguished from noise through their statistical characteristics, different spectral modes can be distinguished and peaks resulting from different hydrometeor types identified (Hildebrand and Sekhon, 1974; Wilfong et al., 1999). Many techniques exist for peak detection to identify when radar spectra are multimodal. Simple options include the identification of noise-floor-separated peaks (Shupe et al., 2004), identifying local minima in the spectral reflectivity (Rambukkange et al., 2011), and skewness signatures (Luke and Kollias, 2013). The MicroARCSL product was developed by Kollias et al. (2007) as a value-added-product for DOE-ARM datasets and was available for the ARM NSA site from 2007-2014. The modality was derived from the spectral data by first identifying Doppler spectral points separated by the noise—floor. Further processing of those spectral points includes identifying local maxima and minima using a 3-dB difference between the relative peaks and valleys. Tools rooted in machine-learning are also useful; Peako (Kalesse et al., 2019) is a supervised algorithm that is first trained on human-identified peaks to then identify peaks in the Doppler spectra. Peaktree (Radenz et al., 2019) is an algorithm that transforms the Doppler spectrum into a binary tree structure, which represents each mode as a

node within the tree. These two tools were combined into a single Peako-Peaktree toolset (Vogl et al., 2024) which facilitates the use of both to identify modes and compute the moments from each mode.

Analyses of cases featuring multi-modal Doppler spectra are useful in understanding when multiple types of hydrometeors are present within a layer and how those scatterers differ from each other. Such cases can facilitate the identification of mixed-phase processes, such as riming indicated by coincident detection of supercooled liquid droplets and snow/ice crystals (Kalesse et al., 2016). In such cases, the cloud liquid droplets and ice particles typically feature different vertical velocities, resulting in distinct peaks in the spectrum (separated by local minima) (cf. the green curve in Fig. 1). Secondary spectral modes can be indicative of secondary ice generation processes, such as Hallett-Mossop rime splintering (Hallett and Mossop, 1974), ice-ice collisional fragmentation (Vardiman, 1978; Takahashi et al., 1995), or droplet shattering upon freezing (Rangno, 2008; Lawson et al., 2017; Korolev and Leisner, 2020). In these cases, the newly generated ice can appear as a slow-falling mode. However, attempts at identification of potentially active processes requires additional information, including polarimetry, temperature profiles, and/or in situ data to understand if there are favorable conditions or necessary ingredients for processes (e.g., riming or rime splintering). Unfortunately, despite the wealth of information they contain, Doppler spectra data are stored in files that many would find prohibitively large to process en masse (e.g., Fabry, 2015) because they contain additional data dimensions compared to the moment data: for example, each gate includes reflectivity, velocity, etc., values in vectors whose lengths corresponds to the number of FFT points (generally 256 or 512). Because of the larger dimensionality, spectra files are much larger than radar moment datasets (approximately 100 MB per hour, ~2.4 GB per day). Because of this, recording Doppler spectra often was limited to specific cases or field campaigns. However, more recent computing and storage systems enable recording and storing Doppler spectra together with the integrated moments for long-term datasets. For example, the DOE/ARM Research Facility (Mather and Voyles, 2013) has collected Doppler spectra for >10 years at the North Slope Alaska and Southern Great Plains sites. Such long-term datasets could allow researchers to better understand detailed microphysical processes like those described above. However, how does one find the metaphorical "needle in a havstack" – the likely small subset of data that are of interest? An objective methodology to efficiently identify these needles amongst the havstacks and effectively extract the microphysical information from the spectra is needed. Further, identifying potential cases of interest using the much smaller moment data files could be advantageous for efficient processing and storage, removing the needs for researchers to download/store enormous amounts of data.

In this study, we propose a methodology for processing vertically pointing radar moment data to identify events with multi-modal Doppler spectra. Relying on radar moment data is advantageous over case-study approaches (e.g., Oue et al., 2018) or by first constraining the dataset temporally to be coincident with thermodynamic observations (e.g., Luke et al., 2021) because of the efficiency by which large numbers of cases may be found. In addition, the method developed here may be applied to multiple sites, whereas most previous studies have only considered single locations. More numerous cases of multi-modal spectra events from different climatic regions may be useful for improving microphysical process identification, and, perhaps, quantification. Further, case studies featuring multi-modal spectra



are more accessible to identify without the need for manual searching of datasets. As such, this novel approach facilitates the study of processes that often lead to multi-modal spectra.

## 2 Data Sources




This study examines data from three cases of cold clouds at three different locations across the United States: (i) Long Island, New York, (ii) Lamont, Oklahoma, and (iii) Utqiagʻvik, Alaska. Each of the sites, outlined below, has a vertically pointing Ka-band polarimetric Doppler radar and routine proximal upper-air observations. Thermodynamic information from these soundings is used to supplement our understanding of each case. Choosing cases from a variety of locations ensures that any signatures associated with multi-modal spectra are robust and occur in both the arctic and in midlatitudes.

## 2.1 Stony Brook (Long Island), New York

The Stony Brook University - Brookhaven National Laboratory Radar Observatory (SBRO), located in Stony Brook, NY (on Long Island), owns and operates the Ka-band Scanning Polarimetric Radar, or KASPR. KASPR is a fully polarimetric radar with high sensitivity and high resolution (Kollias et al., 2014; Kumjian et al., 2020; Oue et al., 2021). The KASPR specifications are available in Table 1. KASPR operations include three scanning strategies: vertically pointing (VPT), plan position indicator (PPI) or surveillance scans, and range height indicator (RHI) scans. Vertically pointing moments and spectra are available approximately every 6 minutes due to the radar cycling between scan types. The vertical resolution of KASPR is 15 m in vertically pointing mode.

Thermodynamic information for this site is obtained through radiosonde launches. Several special launches were made at SBRO to coincide with the Investigation of Microphysics and Precipitation for Atlantic Coast-Threatening Snowstorms (IMPACTS; McMurdie et al., 2022) field campaign IOPs. Additionally, the SBRO site is located 21.4 km away from the National Weather Service in Upton, NY (OKX), which launches operational radiosondes at 11 and 23 UTC daily (valid at 12 and 00 UTC, respectively).

## 2.2 North Slope of Alaska (Utqiavik, Alaska)

The North Slope of Alaska (NSA) research observatory in Utqiagʻvik, Alaska, is operated by the U.S. Department of Energy (DOE). This site has been operational for over 20 years, collecting data with a wide range of instruments ranging from remote-sensing platforms including radars and lidars, to surface meteorological instrumentation and a multi-angle snowflake camera (MASC) (Mather and Voyles, 2013; Stuefer and Bailey 2016; Kollias et al. 2020; Kyrouac and Tuftedal, 2024). In 2011, the DOE Atmospheric Radiation Measurement (ARM) program installed the Ka-band ARM Zenith-pointing Radar (KAZR), a vertically pointing polarimetric Doppler radar at the NSA site (Kollias et al. 2007, Widener et al., 2012; Bharadwaj, 2013; Kollias et al. 2021; Feng et al., 2023). KAZR transmits horizontally polarized waves and receives both horizontal and vertically polarized signals, thereby allowing it to

**Table 1.** Specifications for the radars used to observe the cases used in this study. Note that for the event studied (18 January 2020), KASPR only transmitted in the horizontal (in the same manner as the KAZRs).

| Specification                | SBRO KASPR              | NSA KAZR                   | SGP KAZR                   |
|------------------------------|-------------------------|----------------------------|----------------------------|
| Frequency                    | 35.29 GHz               | 34.86 GHz                  | 34.86 GHz                  |
| Wavelength                   | 8.5 mm                  | 8.6 mm                     | 8.6 mm                     |
| Peak transmit power          | $2.2~\mathrm{kW}$       | $0.2~\mathrm{kW}$          | $0.2~\mathrm{kW}$          |
| Pulse repetition frequency   | $9.92~\mathrm{kHz}$     | $2.77~\mathrm{kHz}$        | $2.77~\mathrm{kHz}$        |
| Transit polarization         | Н                       | H                          | Н                          |
| Receiver polarization        | Simultaneous H, V       | H, V                       | H, V                       |
| Antenna diameter             | 1.2 m                   | 2.0 m                      | 3.0 m                      |
| Antenna Beamwidth            | $0.32^{\circ}$          | $0.31^{\circ}$             | $0.19^{\circ}$             |
| Antenna Gain                 | $53.3~\mathrm{dB}$      | $53.4~\mathrm{dB}$         | $53.5~\mathrm{dB}$         |
| Cross-polarization isolation | -27  dB                 | $-27~\mathrm{dB}$          | $-27~\mathrm{dB}$          |
| Gate spacing                 | 15 m                    | $30 \mathrm{m}$            | $30 \mathrm{m}$            |
| Maximum Range                | $13.5~\mathrm{km}$      | $15\text{-}20~\mathrm{km}$ | $15\text{-}20~\mathrm{km}$ |
| Sensitivity at 1 km          | $-40~\mathrm{dB}$       | $-44~\mathrm{dB}$          | $-48~\mathrm{dB}$          |
| Integration Time             | 1.0 s                   | $3.7 \mathrm{\ s}$         | $3.7 \mathrm{\ s}$         |
| Number of FFT Points         | 1024                    | 256                        | 256                        |
| Nyquist velocity             | $21.06~{\rm m~s^{-1}}$  | $5.87~\mathrm{m~s^{-1}}$   | $5.96~\mathrm{m~s^{-1}}$   |
| Velocity Bin Width           | $0.0412~{\rm m~s^{-1}}$ | $0.0468 \text{ m s}^{-1}$  | $0.0466~{\rm m~s^{-1}}$    |

record the spectra of both the co- and cross-polar signals. The NSA KAZR specifications are in Table 1. KAZR has coarser vertical resolution than KASPR (30 m compared to 15 m; see Table 1). KAZR is collocated with radiosonde stations capable of upper-air observations at the NSA observatory. These soundings are taken twice daily, at 0530 and 1730 UTC.

#### 2.3 Southern Great Plains (Lamont, Oklahoma)

The Southern Great Plains (SGP) atmospheric observatory is located in central Oklahoma, near Lamont, and also is operated by the U.S. DOE ARM program, similar to the NSA site discussed above. Because this site is owned and operated by the same team as the NSA site, the details of the KAZR radar and upper-air observations are similar to those outlined above for the NSA site; the specifications are located in Table 1. Soundings are taken twice daily, at 0530 and 1730 UTC.

#### 3 Signatures of Multi-Modal Spectra





#### 3.1 Identifying Foundational Multi-Modal Cases

Case selection can affect an algorithm's success in detecting secondary modes. There is natural variability in secondary mode events depending on the active microphysical processes, environment, and radar sensitivity. To address this, we select cases from three different vertically pointing Ka-band radars across the United States described in the previous section. We focus our preliminary study on cold clouds, but our methods can be extended to warm-cloud regimes, as well. For each of these "foundational" snow cases, we incorporate three analysis times separated by >2 minutes. In total, we incorporate nine analysis times across three regions to determine if the multi-modal layers have consistent, identifiable signals. The SBRO case comes from the first IOP of the 2020 IMPACTS field campaign on 18 January 2020. A manual analysis of this case indicated there were bi-modal spectra near 5 km ARL from 1855-1908 UTC. The NSA case comes from 7 December 2013, and was selected because it has been explored in detail by Oue et al. (2015). Their study confirmed that bi-modal spectra were present in spectragraphs from 1521 to 1537 UTC. Further, they analyzed the spectral linear depolarization ratio and determined that the secondary mode was likely attributable to columnar ice crystals originating from secondary ice processes (i.e., rime splintering). The SGP case was chosen by manually searching for and evaluating spectra collected during winter months. A case with bi-modal spectra was identified on 13 December 2020 from 1245-1252 UTC. We evaluated these cases for liquid present to provide context into the potential sources of multi-modality (not shown). Using the microwave radiometer at the DOE-ARM sites, both the NSA and SGP cases were noted to have liquid water present in the observed clouds. The NSA case had a liquid water path just under 200 g m<sup>-2</sup> and the SGP case had a much larger liquid water path of over 2000 g m<sup>-2</sup>.

#### 3.2 Doppler Spectral Signatures

The key radar presentation of multi-modal spectra is a broadening of the spectrum across a range of velocities with secondary modes distinctly separated from the primary mode. As such, spectral power or spectral reflectivity in velocity bins between the two modes should decrease to a relative minimum by a measurable threshold (i.e., at least 5 dB). Secondary modes are often observed on the slow-fall-speed side of the primary mode. As such, these secondary modes may result from microphysical processes including primary ice generation, secondary ice production, or the formation of small liquid droplets (cloud or drizzle). The velocity of the secondary mode may change independently 175 from the primary mode; often, the secondary mode's characteristic velocity becomes more negative towards the ground as the fall speed of the growing hydrometeors increases. In many situations, the secondary mode eventually merges with the primary mode. Figure 2 depicts the spectrograms at three times from each of the three multimodal spectra foundational cases described above. The selected times show the same secondary mode evolving over time periods 180 ranging from 12-16 minutes. We examine the secondary mode at different times such that we can capture its natural variations and evolution. Using multiple examples and stages of secondary modes help develop our understanding of how to detect such modes. The selected times shown for each case are separated by at least two minutes and contain multi-modal spectra in layers at least 0.5-km deep, with the secondary mode remaining distinct from the primary mode in that layer.






Across the three cases, the spectrograms reveal broad similarities. The makeup of the primary modes in Fig. 2 can be inferred from their downward velocities, which are related to the particles' fall speeds. The primary modes from the SBRO case in Fig. 2(a i-iii) are suspected to be snow aggregates, given temperatures < 0 °C for the depth of the profile (shown in Fig. 3), dendritic growth zone temperatures from 3.8 to 5.5 km, and downward velocities of about 1.5 m s<sup>-1</sup> (e.g., Locatelli and Hobbs, 1973; Dunnavan, 2021). The hydrometeor type(s) responsible secondary modes, however, are more ambiguous and require further information. However, we do see that, generally, they are associated with slower fall speeds (ranging from 0 to 1 m s<sup>-1</sup>), and thus are inferred to be smaller particles. All secondary modes shown in Fig. 2 display an increase in downward velocity magnitude as they approach the surface, suggesting particle growth. Further, most secondary modes reconnect with the primary mode before reaching the lowest radar sampling altitude. A secondary mode in the NSA case at 1521 UTC at 2.5 km (Fig. 2bi) is disconnected from both the primary mode, and the lower-altitude secondary mode and the primary mode. At later times (not shown), this layer exhibits signs of turbulence, which may contribute to the mode's formation and/or disappearance.

Although each case features broad similarities, each of the cases reveals some subtle differences when examined in detail and across multiple scan times. The SBRO case (Fig. 2a i-iii) reveals a secondary mode (centered at about 4.5-5 km ARL) evolving from a less distinct state characterized by smaller spectral reflectivity and less separation between the primary and secondary modes, to a more distinct state with greater spectral reflectivity values and a greater gap in reflectivity between the primary and secondary modes. In other words, over the 13 minutes shown, the spectral reflectivity of the secondary mode increases by  $\sim 10$  dB as the mode matures. At all analysis times, the secondary mode connects to the primary mode near 4.5 km.

The NSA case (Fig. 2 b i-iii) reveals the greatest temporal variation in its secondary mode. At 1521 UTC (Fig. 2 b i), the spectral modes above and below 2.2 km are disconnected, and a small layer of tri-modal spectra occurs near 2.2 km with otherwise bi-modal spectra above 2.25 km and below 2.1 km. At 1531 UTC (Fig. 2 b ii), the secondary mode centered on 2.5 km has a much smaller spectral reflectivity, and is less distinct from the primary mode. At this same time, the primary and secondary modes extending 1.5-2 km experience a greater separation in the velocity bins of each mode; the slower-falling mode has a velocity near 0 m s<sup>-1</sup> at 2 km, which grows to -0.5 m s<sup>-1</sup> at 1.5 km ARL. By 1537 UTC (Fig. 2 b iii), the spectra are affected by turbulence as inferred from the narrow layers of significantly enhanced spectral widths at 1.5 km and 2.5 km. This turbulence cuts through the secondary mode present from 1-2 km ARL, though the mode is distinct above and below this turbulent layer.

The secondary mode in the SGP case (Fig. 2 c i-iii) is relatively consistent with time, maintaining a similar spectral reflectivity values > -10 dB centered on velocity bins ranging from -0.1 to -0.8 m s<sup>-1</sup>, and retains a similar shape and height throughout the three scans. The secondary mode shifts towards greater fall speeds as it approaches the surface, and merges with the primary mode near 1.5 km. The primary mode sits along the -1.1 to -1.2 m s<sup>-1</sup> velocity

Figure 2. Instantaneous Doppler spectrographs of spectral reflectivity (Z, in dB per velocity bin) for three times for each of the three cases. Spectra from SBRO on 18 January 2020 (a) at (i) 1855 UTC, (ii) 1902 UTC, and (iii) 1908 UTC. Spectra from NSA on 7 December 2013 (b) at (i) 1521 UTC, (ii) 1531 UTC, and (iii) 1537 UTC. Spectra from SGP on 13 December 2020 (c) at (i) 1245 UTC, (ii) 1250 UTC, and (iii) 1252 UTC.

bin at 2 km ARL, becoming -1.5 m s<sup>-1</sup> near 0.5 km ARL. The consistency of these features through the time period shown indicates that their governing physical processes are also persistent throughout this time.

In addition to the similar secondary mode characteristics, all cases presented have another feature in common: turbulent layers. These layers are visibly marked by approximately symmetric, shallow horizontal spike-like features in the spectrograms, extending across a large range of velocities. The turbulent layers in these cases differ in strength, however. Examples of stronger, more well-defined turbulent layers include the NSA case at 1521 UTC (Fig. 2bi) near 3.5 km and at 1537 UTC (Fig. 2 biii) near 2.5 km. The SBRO and SGP cases are riddled with frequent turbulence signatures, including the SBRO case at 1902 UTC (Fig. 2 a-i) at 2.4 km, and in the SGP case ranging from 2.5 – 3 km at all three times shown (Fig. 2 c i-iii).

Given that both the multi-modal spectra and these turbulent layers feature wider spectra spanning a broad range of velocity bins, we seek additional information from the integrated moments to help distinguish between these two types of layers.

### 3.3 Moment Signatures

How do these multi-modal spectra appear in the integrated radar moments? For each of the cases, we examine the moments, of spectrum width (SW) and mean Doppler velocity (MDV). In principle, these two variables should be helpful in classifying and identifying these layers. As mentioned above, SW increases when the Doppler spectrum broadens, either through turbulence or by having multiple, separated modes. The appearance (or disappearance) of a secondary spectral mode could also cause a shift in the MDV. For example, the sudden appearance of smaller, slower-falling particles amongst a background of larger, faster-falling particles could lead to a decrease in the magnitude of the observed MDV (e.g., Schrom and Kumjian, 2016), or that the faster-falling particles have been advected out of the radar sampling volume, etc. Thus, we hypothesize that use of both measurements could prove helpful in identifying multi-modal spectra.

Figure 3 a i-iv shows the temperature profile and radar moments from the SBRO case. The secondary mode layer from 4-5 km ARL has temperatures ranging from -11 to -14 °C, within the dendritic growth zone (e.g., Bailey and Hallett, 2009). The increase in equivalent radar reflectivity factor (hereafter Z; Fig. 3b) towards the ground is consistent with snow particle aggregation, and the MDV (Fig. 3c) near -1.5 m s<sup>-1</sup> is consistent with snow aggregates (e.g., Locatelli and Hobbs, 1973; Dunnavan, 2021). In Fig. 3 a iv, there are three layers of SW greater than the background values (> 0.2 m s<sup>-1</sup>) that coincide with the secondary mode layer and turbulent layers (Fig. 2 a-c) discussed above.

Similar to the SBRO case, the NSA case (Fig. 3 b i-iv) includes numerous layers of SW > 0.2 m s<sup>-1</sup>, some of which can clearly be attributed to turbulence (cf. Fig. 2 b i-ii: at 1521 UTC near 3.5 km and at 1537 UTC near 2.5 km). The SGP case is the warmest of the three, with temperatures near -3 to -5  $^{\circ}$ C (Fig. 3 i). Again, there are numerous regions of SW up to 0.4 m s<sup>-1</sup>, consistent with layers of turbulence identified in the spectrograms, in

Figure 3. Upper-air soundings and vertically pointing radar moment data for each of the three cases. One hour of data for each case is shown. For each case, panel (i) is sounding data with temperature in black and dewpoint in blue; panel (ii) is reflectivity; panel (iii) is mean Doppler velocity; panel (iv) is spectrum width. Note that in (a) the radar plots have data gaps because KASPR switches between scanning strategies and vertically pointing scans are not continuous across the analysis period.

addition to a broader swath of enhanced SW ( $\sim 0.25 \text{ m s}^{-1}$ ) at 1230-1310 UTC at the same height (2 km) as the observed secondary mode (Fig. 3c ii-iv).

Complementary data are valuable in elucidating the underlying processes explaining e features observed on radar. The NSA case has been previously well examined in Oue et al. (2015). That study indicated multiple embedded liquid layers within the cloud at the time immediately prior to our analysis. Both ARM sites (NSA and SGP) have a microwave radiometer (MWR) collocated with the KAZR radars (Cadeddu et al., 2013). From the MWR, the total liquid water and water vapor in the column can be observed (Fig. 4). Both cases have signals indicative of the presence of liquid water within the system that may contribute to either multi-modal signals.

Knowing the times and heights of the multi-modal layers identified from the spectra (Fig. 2), the impacts on the moments in Fig. 3 are apparent. To determine and quantify the relevant signatures that may be used to identify multi-modal layers through radar moments, we manually classify layers in the spectrograms (Fig. 2) as containing multi-modal spectra, turbulence-induced broadening, and neither ("control"). By categorizing the layers, we can quantitatively examine the differences between them as observed in the moment data.

## 3.4 Layer Classification

255

260

The three layer classifications (multi-modal, turbulent, and control) serve to provide a sample of how these features appear in the SW and MDV parameter space and to determine if there is a region in the parameter space specific to one type of layer. Each layer must have a minimum depth of 0.2 km to ensure a sufficient number of data points. When possible, the bounds of the layers are chosen to remain constant across a test case to capture the continuity of the potential processes in that layer, though in some instances they may vary slightly between scans.

## 3.4.1 Multi-Modal Classification

To identify multi-modal layers, we utilize a Bayesian Gaussian Mixture model (GMM) to first detect the number of peaks at every height in the three foundational cases at the nine specified times, using SciKit-learn (Pedregosa et al., 2011). This is done to temporally averaged spectra over ~12 seconds (11.08 s or 3 time steps for both cases using KAZR and 12.39 s or 12 time steps for KASPR) because of the noisy nature of the instantaneous spectra. Additionally, these results are smoothed across a 300-m window (20 gates for SBRO, 10 gates for NSA and SGP) to select the most frequently occurring value for the number of modes within that window. This is done because some gates were outliers in the number of modes detected. This averaging reduces superfluous peaks detected by the automatic peak detection algorithm and mitigates noise caused by rapid switching between two, three, or more peaks when examining the detected peak count with height. The results are shown in Figure 5.

Figure 4. Liquid water and water vapor contents of the two cases located at DOE-ARM sites with a microwave radiometer (MWR) that measures both the liquid water and water vapor along the line of sight path. (a) 7 December 2013 at the NSA site, (b) 13 December 2020 at the SGP site.

Figure 5. (a) Bayesian GMM fit mode count for the foundational cases. Plotted is the average mode count over a 300-m window in height to reduce noise. (b) As in (a), but with the manual layer classification overlaid. Multi-modal layers are designated with purple and uni-modal layers with green. Additionally, turbulent layers are indicated with yellow.

## 3.4.2 Manual Layer Classification





Informed by the results above, we consider manually classified layers in each of the foundational cases to compare the signals that are associated with multi-modal layers and those that are associated with turbulence-induced broadening and control layers. Multi-modal layers are defined to align with the points detected through the Bayesian GMM fitting in section 3.4.1 as well as being confirmed to contain more than one mode by visually examining the spectra. A turbulent layer is defined as encompassing the heights that contain turbulence, visible in the spectra as horizontal spikes in the spectra spanning a large range of velocities (Fig. 2), as well as in the moment MDV as oscillations in relative minimum and maximum MDV values and local maxima of SW (Fig. 3). A control layer is defined as an unambiguously simple layer, containing only a single mode and not containing turbulence. The control layers coincide with unimodal layers from the Bayesian GMM analysis.

As a caveat to these classifications, nature is not always going to cleanly fit into strictly defined boxes, and there are transitions between layer types that are less clearly defined (evident from the Bayesian GMM analysis which often showed individual heights with different amounts of modes than the surrounding layer). Turbulent and control layers are defined by visual inspection of spectragraphs and moment data rather than automatically with pre-defined explicit thresholds because of the large variability for what may be turbulent or "simple" at each date, time, and location. Those quantities are examined after layer definition to determine what patterns exist and how those may aid in the classification and detection of these layers. Across the three times of each case, we can identify a total of 9 layers of turbulence, 16 layers of secondary modes, and 7 control layers (Fig. 6). To examine the differences between these classified layers, we consider vertical profiles averaged across a 145-s period from each of the cases, centered on the times listed. This length of time is chosen because it is the duration of KASPR VPT scans before RHI and PPI scans in the employed scanning sequence. Thus, the data from the NSA and SGP KAZRs is partitioned into 145-s periods to yield a fair comparison.

Layers highlighted in purple in Fig. 6 represent multi-modal spectra. One of the common characteristics of such a multi-modal layer is spectral broadening. Spectrum width alone cannot identify a secondary mode, because turbulent layers see a similar, sharper spike in spectrum width. However, as seen by the cyan error bars in Fig. 6, the standard deviation of mean Doppler velocity, hereafter  $\sigma(MDV)$ , in a turbulent layer is quite different from that in a secondary mode. Turbulent layers feature large  $\sigma(MDV)$  associated with the variable vertical fall speeds induced by turbulence, whereas the secondary modes have low  $\sigma(MDV)$ . A large variance in mean Doppler velocity over the 145-s period can be used to detect highly turbulent layers and eliminate them from being marked as potential secondary modes.

## 4 Establishing the Criteria and Detection Methodology

In aggregate, we use 781 data points to determine the typical values associated with each layer. The contributions
from each site to each layer type are shown in Table 2. The SBRO case has disproportionally more points due to
KASPR's finer vertical resolution. We examine the average value for each layer type by case in addition to an overall

Figure 6. Radar moment MDV and SW for three times for each of the three cases: (a) SBRO at (i-ii) 1855 UTC, (iii-iv) 1902 UTC, and (v-vi) 1908 UTC; (b) NSA at (i-ii) 1521 UTC, (iii-iv) 1531 UTC, and (v-vi) 1537 UTC; (c) SGP at (i-ii) 1245 UTC, (iii-iv) 1248 UTC, and (v-vi) 1252 UTC. Purple shading indicates multi-modal layers, yellow is turbulence, and cool green are control layers, unaffected by spectrum-broadening processes. Pink and cyan error bars along each black line is the standard deviation of each moment variable, taken in time over 145-s periods.

average (Table 3) to ensure that the uneven number of samples from each radar/site does not bias the statistics. The results of this analysis reveal that individual variables are not distinct between the three categories: both the multi-modal and turbulent layers have a near identical  $\overline{SW}$ , whereas both the multi-modal and control layers have a near identical average  $\sigma(MDV)$ . The joint distribution of these variables, however, makes clear the distinctions between these three types of layers and allow us to separate the three categories into distinct portions of the parameter space (Fig. 7). Layers containing a secondary mode (purple) occupy the bottom right of the parameter space in Fig. 7, marked by large  $\overline{SW}$  (> 0.17 m s<sup>-1</sup>) but small  $\sigma(MDV)$  (< 0.1 m s<sup>-1</sup>). Although some overlap exists with the control (green) and turbulence (yellow) points, these make up a small portion of the total points. Otherwise, there is little overlap with the other layer classifications. Turbulence-containing layers (yellow) only present in the upper region of the parameter space, where  $\sigma(MDV) > 0.1$  m s<sup>-1</sup>. Control layers (purple) are confined to the bottom left portion of the parameter space, with  $\sigma(MDV) < 0.1$  m s<sup>-1</sup> and  $\overline{SW} < 0.17$  m s<sup>-1</sup>. Given this observed separation in the  $\overline{SW}$ -  $\sigma(MDV)$  parameter space from known cases in different regions observed with different radars, we hypothesize that these variables generally can be used to determine the presence of a secondary spectral mode in the moment data. To proceed with testing our hypothesis, we will use  $\overline{SW} > 0.17$  m s<sup>-1</sup> and  $\sigma(MDV) < 0.1$  m s<sup>-1</sup> as the criteria to detect multi-modal layers observed in snow cases with vertically pointing Ka-band Doppler radar.

**Table 2.** Total number of data points contained in each layer type and each case.






| Condition    | Total Data Points | SBRO Layer | NSA Layer | SGP Layer |
|--------------|-------------------|------------|-----------|-----------|
| Control      | 101               | 62         | 21        | 18        |
| Mode         | 542               | 407        | 84        | 51        |
| Turbulence   | 138               | 94         | 21        | 23        |
| Total Points | 781               | 563        | 126       | 92        |

The method for applying these criteria is shown schematically in Fig. 8. The first step is to create time-averaged vertical profiles of MDV and SW in the same manner described in section 3.4 (i.e., 145-s segments). Next, specific height levels with time-averaged  $\overline{SW} > 0.17 \text{ m s}^{-1}$  and  $\sigma(MDV) < 0.1 \text{ m s}^{-1}$  are flagged, herein considered "flagged points."

Two additional filters are needed to avoid detecting non-events: a noise filter and a rain filter. If there is low signal-to-noise ratio (SNR), data may be unreliable. Any flagged points with SNR < -5 dB are excluded. Liquid precipitation such as rain causes the Doppler spectra to broaden, as the wide variety of sizes of droplets will have a wide range of associated fall speeds. To maintain a focus on drizzle and ice processes, we use the rate of change of LDR with height to identify the melting layer and exclude data between the melting layer and the surface at that time. Points with a rate of change of LDR exceeding -0.02 dB/m are identified as the top of the melting layer and determined to be coincident with a SNR > 1 dB and downward velocity of at least 1 m s<sup>-1</sup> to avoid detecting cloud ice, which was a potential concern. This filter reduces the impact of spectral broadening due to rain (the

Figure 7. (a) Comparison of uni-modal and multi-modal layers by two methods - uni- and multimodal parameters by case plotted with error bars, with their aggregate average plotted by stars. The markers at the center of the cross-hairs represent the average  $\overline{SW}$  and  $\sigma(MDV)$  for the set of points in each category. The extent of the cross-hairs in each direction represents the standard 7 deviation  $\overline{SW}$  and  $\sigma(MDV)$  of for the set of points in each category. Similarly (b) contains the same results with values restricted to  $\sigma(MDV) 

Figure 8. Flow chart explaining the procedures followed when applying the multi-modal detection algorithm. Input VPT moment data are averaged into vertical profiles and points meeting the specified thresholds are identified and flagged. Points with too low of signal or suspected to be associated with rain are then eliminated. Flagged points are then further processed to determine the number occurring per hour, and the LDR associated with each flagged point is saved in association with the event. Finally, if the hourly flag total exceeds the threshold of 100 flags per hour, a case is identified.

Table 3. Average values of mean spectrum width  $(\overline{SW})$  and the standard deviation of mean Doppler velocity  $(\sigma(MDV))$  for control, mode, and turbulence layer types. Averages are taken across each case and across all cases combined. All values shown are in units of m s<sup>-1</sup>. Note that the individual case values represent that there is variability associated with the unique aspects of individual cases in different environments, and that the total values that average the cases together are more representative of the general values associated with each layer type.

| Parameter  |                   | Total<br>Avg | Total Std. Dev. | SBRO  | NSA   | SGP   |
|------------|-------------------|--------------|-----------------|-------|-------|-------|
| Control    | SW                | 0.164        | 0.027           | 0.180 | 0.124 | 0.165 |
|            | $\sigma_{ m MDV}$ | 0.112        | 0.042           | 0.131 | 0.105 | 0.058 |
| Mode       | SW                | 0.198        | 0.049           | 0.191 | 0.244 | 0.184 |
|            | $\sigma_{ m MDV}$ | 0.061        | 0.025           | 0.063 | 0.061 | 0.049 |
| Turbulence | SW                | 0.207        | 0.049           | 0.198 | 0.242 | 0.205 |
|            | $\sigma_{ m MDV}$ | 0.178        | 0.077           | 0.205 | 0.150 | 0.086 |

wider distribution of droplet sizes leads to an enhancement of spectrum width that impairs the ability to detect a secondary mode) and is advantageous over using larger fall speeds alone to detect rain because that may also exclude situations where large (>2-mm diameter) graupel (e.g., Lamb and Verlinde, 2011; Heymsfield et al., 2018) dominates the backscattering.



To mitigate noise associated with false detections and to focus on spatiotemporally consistent features that are likely of more microphysical relevance, we examine how many points are flagged in specified time periods (e.g., 1 hour). Periods with large counts of flagged points ("flag counts") can then be considered for further analysis, such as microphysical process determination. Because the flag count is determined by the number of points meeting the multi-modal spectral detection criteria in a one-hour period, large flag counts could arise from thick layers and/or persistent signals. When examining the foundational cases at the SBRO, NSA, and SGP sites, there are 33-143 flags per 145-s segment during segments with known multi-modal layers<sup>1</sup> excluding the 1855 UTC SBRO scan. To get an idea of the expected flag counts in one-hour periods, we applied the flagging algorithm to a full hour of continuous VPT data from the KAZR cases (containing the known multi-modal spectra), and found 1201 flags from 12-13 UTC in the SGP case, and 1541 flags from 15-16 UTC in the NSA case. Based on these counts, we set the threshold for a large hourly flag count at 100 flags per hour. A 100-flag hr-1 criterion should be sufficient to capture thinner, sustained layers in addition to deeper layers. Note that, depending on one's application (i.e., studying drizzle

<sup>&</sup>lt;sup>1</sup>This choice is made due to the specific manner KASPR is run, the VPT scans run for only 145s. The standard deviation values are dependent on the durations used. Generally, if one wishes to repeat this on other datasets, we recommend using time segments close to 2-2.5 minutes.

formation, secondary ice production, etc.), the period over which flagged data points are counted and/or the flag count threshold may be adjusted, as needed, to address the timescale on which those processes are observed.

As an example of how this detection algorithm can be used, it was run on the cases used to build it. While this clearly is not an independent test, it serves to demonstrate the results of the algorithm and explore how well the test cases meet their own criteria. For the times and heights that were identified as containing multi-modal spectra, all three cases display gates that are flagged for meeting the criteria (Fig. 9). However, there are times for which the criteria are not met. For example, the 1855 scan at the SBRO site (Fig. 9a) features no points that meet the multi-modal layer criteria, despite the appearance of the spectra (see Fig. 2 a i) and the Bayesian GMM mode detection. The 1855 UTC scan is the first to appear multi-modal; Bayesian GMM is sensitive enough to detect this multimodality, whereas our methodology using radar moment data does not because the spectra did not have  $\sigma(MDV) 

Figure 9. Results of the detection algorithm applied onto the foundational cases for (a) SBRO (b) NSA (c) SGP. In (a) the flags are indicated by thin horizontal lines because the VPT data is only available in the short time periods used to compute the flags. (b) and (c) are shaded and continuous in time because KAZR operates continuously in VPT mode. Pink indicates presence of only the flag with  $\overline{SW} > \text{m s}^{-1}$  and  $\sigma(MDV) > 0.1 \text{ m s}^{-1}$ ), blue indicates presence of only the flag with  $\sigma(MDV) < 0.1 \text{ m s}^{-1}$  and  $\overline{SW} < 0.17 \text{ m s}^{-1}$ , and black indicates all necessary criteria for a flag being met. Red dotted lines indicate the maximum bounds in time and height used to define the multi-modal layers used to obtain the criteria.

km at 1225 UTC and descending over time until 1300 UTC, and a second starting near 2.4 km at 1250 UTC and descending over time beyond 1325 UTC.

#### 5 Conclusions



Outcomes and Conclusions Through examining three case studies, consistent features in radar moment data were found to be characteristic of multi-modal spectra. When examining vertically pointing data for 145-s periods, we find that multi-modal layers have a relatively large mean spectrum width  $\overline{SW}$  (> 0.17 m s<sup>-1</sup>) and relatively low standard deviation of mean Doppler velocity  $\sigma(MDV)$  (< 0.1 m s<sup>-1</sup>), occupying a distinct section of the  $\overline{SW}$ -  $\sigma(MDV)$  parameter space different than that of turbulent and control layers. These features were quantified such that they can be used in an event detection algorithm. By identifying similar layers consistent with the multi-modal signals, we can identify likely multi-modal spectra through moment data alone, without having to process the complete radar spectra files.

For this reason, we hypothesize that  $\overline{SW}$  and  $\sigma(MDV)$  can be used to determine the presence of a secondary mode in radar moment data. Although the preliminary testing done here is encouraging, to robustly test this hypothesis

requires evaluating these criteria for a much larger, independent dataset. This requires a long-term radar dataset, such as that available from the NSA site. The success of the criteria in detecting multi-modal layers using the moment data exclusively will be evaluated, with verification by manual analysis of the Doppler spectra, in Part 2.


Development of these criteria into an algorithm for event detection will also allow for efficient processing of long-term radar datasets, which opens the possibilities of creating a climatology of multi-modal spectra events. Additional analysis on detected cases may open the doors to process identification and determining if modes are composed of ice or liquid droplets, using additional observations.

The design of the criteria and methodology is targeted at reducing false positives; it is likely that the use of these criteria may miss the detection of some multi-modal spectra (cf. Fig. 7). This may be due to modes having weak power returns or little separation in velocity bins from the primary mode. The criteria we determined herein can be adjusted by those applying this detection algorithm to other datasets. However, relaxing the criteria will have trade-offs. By decreasing the  $\overline{SW}$  threshold, one will identify more secondary modes that may be less separated from the primary mode, but may also begin to falsely identify single-mode layers that would need to be manually identified from spectra files and removed from the results. Similarly, relaxing the criteria to include larger values of  $\sigma(MDV)$  may result in turbulence being mistaken for a secondary mode. Applying these criteria to datasets with significantly different radar systems may require additional adjustments. Nonetheless, the classification technique using objective criteria can help to analyze characteristics of turbulent layers and their role in microphysics and snow intensification using more than case studies (e.g., Oue et al., 2024). We propose that this approach can be applied for efficient detection of multi-modal Doppler spectra in large datasets, as we show in Part 2.

Code and data availability. KAZR moment and spectra data are available online at https://adc.arm.gov/discovery.

Author contributions. This work stems from the dissertation work of SW, advised and guided by MRK. MO and PK provided KASPR data and insights into its use.

Competing interests. One of the authors is a member of the editorial board of the journal "Atmospheric Measurement Techniques".

Acknowledgements. Support for the authors comes from the U.S. Department of Energy Atmospheric System Research program Grant DE-SC0018933, and from academic appointments at The Pennsylvania State University. Data were obtained from the Atmospheric Radiation Measurement (ARM) Program sponsored by the U.S. Department of Energy, Office of Science, Office

| 430 | of Biological and Environmental Research, Climate and Environmental Sciences Division. We are grateful for the DOE ARM program's continued support of the KAZR facilities, data processing, and data storage that made this study possible. |  |  |  |  |
|-----|---------------------------------------------------------------------------------------------------------------------------------------------------------------------------------------------------------------------------------------------|--|--|--|--|
|     |                                                                                                                                                                                                                                             |  |  |  |  |
|     |                                                                                                                                                                                                                                             |  |  |  |  |
|     |                                                                                                                                                                                                                                             |  |  |  |  |

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
