# Peer review of "Detection of Multi-Modal Doppler Spectra. Part 1: Establishing Characteristic Signals in Radar Moment Data"

_EGUsphere, 2025_

## Author Comment (AC1)

Reviewer #1:

**Major Comments:**

Major Comment 1: The authors should come up with a better way of quantifying multi-modal spectra. The authors could easily develop quantitative metrics for degree of multi-modality using a variety of techniques such as fitting the spectra at each height with a Gaussian Mixture model and by using the Gaussian mixture components to derive relevant statistical measures of multi-modality. My concern is that the authors are being a bit too selective about what they consider to be multi-modal and unimodal. An objective metric that is more independent of researcher biases here would be more informative and would provide a fairer representation of multi-modal/unimodality inherent in the spectra data.

Thanks for this feedback. This also came up as a similar suggestion from Reviewer 2 to consider a peak detection toolset similar to the Gaussian Mixture model (abbreviated GMM going forward). Ultimately, we have followed your recommendation and now utilize a Bayesian Gaussian Mixture model to first detect the number of peaks at every height in the original three cases at the nine specified times. This is done to temporally averaged spectra over approximately 12 seconds (with the varied temporal resolution of KAZR and KASPR, this is more precisely 11.08 s (3 time steps) for both cases using KAZR and 12.39 s (12 time steps) for KASPR). This averaging reduces the superfluous peaks detected by the automatic peak detection function and noise generated by rapidly switching between 2, 3, or greater peaks when observed the detected peak count with respect to height.

We now use the results of this to inform layer selection of "control" and "mode" layers, and those previously selected layers are adjusted in the revision to confirm their uni-modal or multi-modal status before assigning them to either a control or mode category. Taking the SBRO case for example, the previous analysis was indeed missing a subtle multi-modal layer that overlaps with the previous "control" layer for that case. While this lower multi-modal layer is much less distinct, it is detectable through spectral analysis. An additional multi-modal layer is added to the analysis and the "control" layer is now represented elsewhere, when a uni-modal layer unaffected by turbulence can be identified. Additionally, the depth of the upper level multi-modal layer is now extended. For the NSA case, the control layers are minorly affected and shortened to accommodate the limited uni-modal depth identified with GMM. For the SGP case, we were missing a low-level multi-modal layer that is now reclassified and the uni-modal control layers were relocated to near the top of the cloud.

[Figure]

Fig. R1: Bayesian GMM fit mode count for the foundational cases. Plotted is the average mode count over a 300-m window in height to reduce noise.

[Figure]

Fig. R2: This is a modification of Fig. R1 with the adjusted layer shading overlaid to indicate how we ensured that only uni-modal layers are able to be designated "control" and only multimodal layers are able to be designated "mode" -- mode is designated with purple and control with green for this example of how the new layers line up, turbulence is indicated with yellow shading. Opacity is related to the count of times that use those heights. For example, the SGP case uses all the same layers for all 3 times and thus the layers combine to alpha=0.3, whereas a layer used by a single height will have alpha=0.1 (a lighter, less opaque shade). These are plotted for each time individually in Figure R3 below in our revision of Figure 5.

To summarize, the main impacts from this new analysis are (1) adding a low-level "mode" layer to SBU (2) relocating the "control" layer for SBU (3) slightly shortening the SGP "mode" layer considering where the bi-modal classification ends (4) removing the "control" layer from the NSA case and carefully relocating it to the non-turbulent area around 3 km. Note that in many cases the uni-modal layer is not continuous (SBRO at 2.6-2.8 km AGL, NSA 1.9 km AGL in the 1531 scan, and others). Some of the ambiguity on the gaussian GMM detecting a uni- or multi-modal comes from (1) the role of turbulence and (2) increase in the noise floor above 30 dB (note the upper levels of NSA and SGP with 0 modes). The improved layer selections will now be represented in Figure 4 of the revision, and below in Figure R3.

[Figure]

Fig. R3: Updates to the previous iteration of figure 4, with corrections and adjusted layer selections. As before the caption will read: "Radar moment MDV and SW for three times for each of the three cases: (a) SBRO at (i-ii) 1855 UTC, (iii-iv) 1902 UTC, and (v-vi) 1908 UTC;

(b) NSA at (i-ii) 1521 UTC, (iii-iv) 1531 UTC, and (v-vi) 1537 UTC; (c) SGP at (i-ii) 1245 UTC, (iii-iv) 1248 UTC, and (v-vi) 1252 UTC. Purple shading indicates multi-modal layers, yellow is turbulence, and cool green are control layers, unaffected by spectrum-broadening processes. Pink and cyan error bars along each black line is the standard deviation of each moment variable, taken in time over 145-s periods."

Shown in Figure R3, we retain the original style of human-influenced layer selection (designated turbulent, mode, control) though this is now supplemented by the analysis of the GMM output (which in figures R1 and R2 designated uni-modal and multi-modal heights). The following figure "Fig. R4" is a replacement for former Figure 5 in the preprint. The former heat-map is removed for visual clarity and instead it now contains only the mean and error bar visualizations. This is provided for several subsets of datapoints, in Fig. R3a we show how the contributing cases fill the parameter space when examined via layer classification and via GMM mode identification. In Fig. R3b, we show the same quantities with restricted stdev(MDV) to represent the effect of the use of that parameter to exclude signals caused by turbulence.

As a result, the parameter space of mean moment SW and standard deviation moment MDV have changed. In the most recent iteration, a multi-modal layer via layer analysis has a mean SW of 0.19 m s$^{-1}$ and a multi-modal layer via the GMM technique has a mean SW of 0.175 m s$^{-1}$. Examining both groups again with likely turbulent layers filtered out (corresponding to a standard deviation of MDV greater than 0.2 m s$^{-1}$ ) multi-modal layer via layer analysis has a mean SW of 0.19 m s$^{-1}$ and a multi-modal layer via the GMM technique has a mean SW of 0.165 m s$^{-1}$. While this is close to our previously indicated criterion of 0.19 m s$^{-1}$, we plan to re-run the results of Part 2 with an adjusted value of 0.17 m s$^{-1}$ applied. We expect this to somewhat increase the total number of cases and case lengths, and to potentially detect cases that do not meet the previous verification criteria as outlined in both initial submissions.

Examining these added figures, it is clear that the GMM analysis is able to discern more subtle modes that do not always meet our criteria for what we considered as "distinct" with a detectable drop in reflectivity between the modes (see line 263 in the preprint of part 1: a 5 dB or larger drop in reflectivity between the modes). This new analysis has given us this additional reference point for the potential for a user to adjust their SW criterion when interested in detecting more subtle secondary and tertiary modes, and for multi-modal events that this criteria may miss.

We have modified the revised text to explain this revised approach and to make the points discussed above regarding our methods and associated with the revised figures.

[Figure]

Fig. R4: (a) Comparison of uni-modal and multi-modal layers by two methods – uni- and multi-modal parameters by case plotted with error bars, with their aggregate average plotted by stars. The markers at the center of the cross-hairs represent the average **_SW_** and **_σ(MDV)_** for the set of points in each category. The extent of the cross-hairs in each direction represents the standard

deviation $SW$ and $\sigma(MDV)$ of for the set of points in each category. Similarly (b) contains the same results with values restricted to stdev(MDV)<0.1 m s$^{-1}$ (our turbulence threshold).

Unshown in this comment, we also tried k-means clustering as an unbiased method to categorize and classify the regions of the parameter space, but opted for the GMM results as they have a mathematical connection to the modality, though we noticed the clustering method also suggested a distinction between categories at mean(SW) near 0.15 m/s.

Major Comment 2: What is the expected miss rate and false positive rate of the detection algorithm? From Figure 7 it seems like there is a missed multi-modal spectra as pointed out in the main text. However, because the authors only look at three cases, it's hard to actually evaluate the performance of the algorithm. The authors will need to be very careful in part 2 to objectively evaluate the algorithm's performance and to properly quantify the hits, misses, false alarms, and nulls using a larger dataset. In my opinion, this really necessitates the ability for the authors to provide a much more objective set of criteria for determining multi-modality or uni-modality.

Much of this is addressed in part 2, the preprint of which is available now at https://doi.org/10.5194/egusphere-2025-672 (with revisions in progress as well). The original algorithm was tested on three years of data from the KAZR at the NSA site, independent of the case in this portion of the paper.

Additionally, this has now been re-run with the adjustments made to the criteria discussed in response to Major Comment #1. This includes adjusting the SW criteria that was responsible for the previous run missing the multi-modal layer at 19:02 UTC in the SBRO case (see below Figure R5 to see improvement). We were also being too cautious with including that lower segment of the multi-modal layer in the 19:02 UTC SBRO observation into the designated layer used for parameter determination (see above with the newly applied GMM approach to improve our layer identification), and that has been adjusted as well within the next version to be submitted.

[Figure]

Fig. R5: Previous figure 7 edited to include modified "mode" zones and with the reduced threshold on SW. Revision caption to read: "Results of the detection algorithm applied onto the foundational cases for (a) SBU (b) NSA (c) SGP. In (a) the flags are indicated by thin horizontal lines because the VPT data is only available in the short time periods used to compute the flags. (b) and (c) are shaded and continuous in time because KAZR operates continuously in VPT mode. Pink indicates presence of only the flag with $SW$>0.17 m s-1 and $\sigma(MDV)$>0.1m/s), blue indicates presence of only the flag with $\sigma(MDV)$ <0.1 m s-1 and $SW$<0.17 m s-1, and black indicates all necessary criteria for a flag being met. Red dotted lines indicate the maximum bounds in time and height used to define the multi-modal layers used to obtain the criteria."

While we referred to the GMM results for guidance, it was also interesting to note that the GMM analysis failed to identify the multi-modal layer between 1.5-2.0 km AGL in the NSA case, which is visually apparent. That layer is detected by our check of the methods on the foundational cases however (above, Fig. R5, new Figure 7). As such, we are concerned that the Gaussian mixture method is not sufficient to use as the sole source of verification.

Considering only the results of the GMM analysis to determine the uni- or multi-modality, we examined the parameter space for all points and for all points that have a stdev(MDV) below the threshold of 0.1 m/s. We examined the ratio of points that fall below and above the mean(SW) of the uni- and multi-modal subsets. With turbulence the mean value was 0.175 and without turbulence the mean value was 0.16; in both analyses approximately 43% of points in the multi-modal category exceeded this. While this would suggest a large expected missed detection, we account for the margin of error in the implementation of the criteria by examining the cases by hour and by height. Existence of a predicted multi-modal case is dependent on (1) large flag counts (2) clustering of the flags within a layer or fall streak and

quantified considering (3) a percentile approach to identify the predicted top and bottom of the layers.

When designing the algorithm, we were tolerant of missed detections (particularly some of those more subtle ones highlighted by the GMM analysis above) in favor of more robust features. For example, in the 1521 NSA case, there is a low reflectivity (-20 dBZ), detached, secondary mode located at 2.5 km AGL that GMM detected but was not robust enough to affect moment SW. Rather, we expect (and have seen within results for part 2) that the technique is more suited for identifying multi-modal cases with somewhat greater reflectivity coming from both modes (as in those seen in this paper at lower levels within the NSA case and those modes near 2 km AGL in the SGP case). Inherently the moment SW is linked to the contributing power (reflectivity), causing the detectable increase in SW used in the criteria. Having a large number of false positives, in our view, would make the searching more time consuming and inefficient. We address this philosophical choice in the revised manuscript, and suggest that readers interested in detecting a higher percentage of all multi-modal features can adjust the parameters accordingly.

Accordingly, with the edits made to the methodology within Part 1 that improve the parameter choices for layer detection, much of this comment is addressed in the edits to Part 2.

--------- --------- --------- --------- --------- --------- --------- --------- --------- --------- ---------

**Minor Comments:**

Minor Comment 1: The authors state in the conclusions that: "The design of the criteria and methodology is targeted at reducing false positives; it is likely that the use of these criteria may miss the detection of some multi-modal spectra (cf. Fig. 7)." This seems backwards to me. If anything, wouldn't you want to reduce missed events? The reasoning is that you can always revisit the spectra data for flagged periods to verify multi-modality after the fact. Therefore, false positives, unless there are many, aren't that big of a deal whereas missed multi-modal spectrums represent a serious deficiency in your algorithm as possibly suggested by Figure 7.

Interesting question -- we suppose it comes down to a matter of philosophy. When designing the algorithm, we were tolerant of missed detections (particularly those more subtle ones highlighted by the GMM analysis above) in favor of more robust features. Having a large number of false positives, in our view, would make the search more time-consuming. We can address this philosophical choice in the revised manuscript and suggest that readers interested in detecting a higher percentage of all multi-modal features can adjust the parameters accordingly. We expect that many of the points that would be missed may be adjacent to previously detected layers and may affect results such as layer depth as discussed in part 2.

Ultimately, a balance of the tolerance for the missed events with tolerance for false positives would be interesting, but the amount of data required to test the missed events makes this prohibitively time consuming and computationally expensive. In our ongoing revisions to part 2, we will be including a subset that was tested for missed events, however with the extreme computational cost of processing the full three year period at the resolution required, it would be prohibitively expensive computationally and in time for the entire verification analysis.

Minor Comment 2: I was disappointed to see a lack of complimentary data from other instruments present at these three sites. The SBU site in particular has a wealth of potentially useful ground instruments like Pluvio, Parsivel, and the MASC imager. These instruments could be helpful in part 1 of this paper to provide some hints to readers as to the actual mechanisms behind periods of multi-modal spectra. I don't necessarily think that the authors have to include such data in their analysis (perhaps that will be in part 2?). However, I do think that the authors should at least consider including one or two figures that use these additional instruments to help provide context into the potential sources of the multi-modality for each of the cases. That is, do MASC images of the particles at the ground support hypotheses of rime splintering, ice-ice collisions, or freezing drop fragmentation near regimes of multi-modal Doppler spectra?

We agree that the complementary data are very valuable in parsing out the details and trying to identify possible active processes. In part 2, following the long-term analysis we then follow that with a detailed look into three detected of the cases. Those cases, all observed at the ARM NSA site, are analyzed in more depth to discuss the possible processes and features being detected by the algorithm. Additionally with all verification cases occurring at the same site, they should have the same available instrumentation (depending on instrument down time and maintenance). Part 2 received similar feedback; as such, we incorporate and will strengthen these types of complementary observations that were available on the dates used in part 2.

Specifically addressing the NSA case: Oue et al. 2015 does address this case in a much more comprehensive manner already, including both the Ka-band KAZR and X-band XSAPR and Microwave Radiometer (MWR). That previous analysis indicated multiple embedded liquid layers within the cloud depth, though cuts off at approximately our analysis time. Below in figure RX, we plot the liquid water path and total water vapor during the periods shown. During the analysis period, as with in Oue et al. 2015, the NSA case has LWP of approximately 200 g/m2, indicating the liquid layers identified in the previous study still exist within this period. The SGP case has much greater values of both LWP and total water vapor detected by the MWR, suggesting the role of liquid water in cloud processes and potentially (but unconfirmed) liquid cloud droplets comprising identified modes.

The SBRO case was a part of the IMPACTS field campaign; while unshown in this paper the P-3 research aircraft did observe liquid cloud droplets at approximately 5 km AGL near the analysis within this time over Long Island, though not flown close enough to directly over KASPR and hence unshown. We did a dual-wavelength ratio analysis of the SBRO case using W-band observations from ROGER which suggested aggregation as a dominant process throughout the times shown and depth of the observations. While MASC images are not a part of the available IMPACTS dataset, the Parsivel disdrometer and Doppler lidar were observing during this event. Per the disdrometer, moderate snowfall was either intermittent or continuous across the period. Particles ranging in size from (on average) 0.25-2 mm were observed falling at 0.5-2.5 m/s, peaking at 0.6 mm and 1 m/s across the analyzed timed. With some smaller, 0.5 mm particles observed at faster speeds near 2 m/s, it is possible that riming could have occurred aloft causing small observed particles to have a large enough mass to result in fast fall speeds (but not conclusive enough to state within our analysis). Lidar indicates likely liquid water near the surface (below 300 m AGL) that could contribute to riming, but does not have signal aloft to determine if higher liquid layers exist.

This commentary about observations of liquid in the foundational cases will be added to the text of section 3.2 Doppler Spectral Signatures and indicate if liquid layers may be present in those cases, contributing to plausible explanations for modes being observed.

[Figure]

Figure R5: A look into the liquid water and water vapor contents of the two cases located at DOE-ARM sites with a microwave radiometer (MWR) that measures both the liquid water and water vapor along the line of sight path. Both cases have signals indicative of the presence of liquid water within the system that may contribute to either multi-modal signals or cloud and precipitation processes.

---

## Author Comment (AC2)

Reviewer #2:

**Major Comments:**

Major Comment 1: In my opinion the detection of a second mode and turbulence is to subjective. You are likely missing a lot of cases due to your rather arbitrary requirement of a 5dB reduction of spectral Ze and a minimum spectral Ze of -20dB for a multi-modality. Especially because your entire technique is based on those few subjective cases. Also, I am not sure if the number of cases you used to establish the MDV and SW thresholds are sufficient. It could very well be that these few cases are special, especially since you had such subjective selection criteria. To my knowledge, there are a few Doppler spectra peak detection algorithms openly available. For example, I have recently used the peako-peaktree toolset (https://amt.copernicus.org/articles/17/6547/2024/) and found it to be easily applicable to Doppler spectra of several different radars. In my opinion, detecting peaks in such an automatic, objective way would greatly benefit your study, especially since then you can use much more data to make your MDV, SW clustering and threshold definition more robust. Similar algorithms are available for turbulence detection, for example eddy dissipation rate retrievals. I would suggest to first apply a peak detection algorithm, filter for cases where multiple peaks were obtained (also perhaps were the feature is consistent enough to be caused by microphysics) and then do the clustering in sigma(MDV) mean(SW) space. If you apply a peak detection algorithm, also your validation of the study is much easier, as you do not need to manually go through a lot of cases in order to visually see if your algorithm is valid. This is also why I think you do not need 2 papers to describe the method. Using a peak detection algorithm already reduces this study to the clustering in sigma(MDV) mean(SW) space, allowing you to cover the validation of the method in the same paper.

Thanks for this feedback, this also came up as a similar suggestion from Reviewer 1 to consider a Gaussian Mixture model approach to doing this (abbreviated GMM going forward). I initially pursued setting up the peako-peaktree toolset as you recommended; however, within the documentation and tutorial scripts provided it was stated that processing one hour of data could exceed 30 minutes, which, while feasible for the initial cases in Part 1, becomes quite computationally expensive if scaled up to a long-term dataset or used for a three-year verification project.

Ultimately, we have followed the previous recommendation for the GMM approach and now utilize a Bayesian Gaussian Mixture model to detect the number of peaks at every height in the original three cases at the nine specified times. This is done to temporally averaged spectra over approximately 12 seconds [with the varied temporal resolution of KAZR and KASPR, this is more precisely 11.08 s (3 time steps) for both cases using KAZR and 12.39 s (12 time steps) for KASPR]. This averaging reduces the superfluous peaks detected by the automatic peak detection function and noise generated by rapidly switching between 2, 3, or greater peaks when observed the detected peak count with respect to height.

We now use the results of this to inform layer selection of "control" and "mode" layers, and those previously selected layers are now adjusted in the revision to confirm their uni-modal or multi-modal status before assigning them to either a control or mode category. Taking the SBRO case for example, the previous analysis was indeed missing a subtle multi-modal layer

that overlaps with the previous "control" layer for that case. While this lower multi-modal layer is much less distinct, it is detectable through spectral analysis. An additional multi-modal layer is added to the analysis and the "control" layer is now represented elsewhere, when a uni-modal layer unaffected by turbulence can be identified. Additionally, the depth of the upper level multi-modal layer is now extended. For the NSA case, the control layers are minorly affected and shortened to accommodate the limited uni-modal depth identified with GMM. For the SGP case, we were missing a low-level multi-modal layer that is now reclassified and the uni-modal control layers were relocated to near the top of the cloud.

[Figure]

Fig. R1: Bayesian GMM fit mode count for the foundational cases. Plotted is the average mode count over a 300-m window in height to reduce noise.

[Figure]

Fig. R2: This is a modification of Fig. R1 with the adjusted layer shading overlaid to indicate how we ensured that only uni-modal layers are able to be designated "control" and only multimodal layers are able to be designated "mode" -- mode is designated with purple and control with green for this example of how the new layers line up, turbulence is indicated with yellow shading. Opacity is related to the count of times that use those heights. For example, the SGP case uses all the same layers for all 3 times and thus the layers combine to alpha=0.3, whereas a layer used by a single height will have alpha=0.1 (a lighter, less opaque shade). These are plotted for each time individually in Figure R3 below in our revision of Figure 5.

To summarize, the main impacts from this new analysis are (1) adding a low-level "mode" layer to SBU (2) relocating the "control" layer for SBU (3) slightly shortening the SGP "mode" layer considering where the bi-modal classification ends (4) removing the "control" layer from the NSA case and carefully relocating it to the non-turbulent area around 3 km. Note that in many cases the detected uni-modal layer is not continuous (SBRO at 2.6-2.8 km AGL, NSA 1.9 km AGL in the 1531 scan, and others). Some of the ambiguity on the gaussian GMM detecting a uni- or multi-modal comes from (1) the role of turbulence and (2) increase in the noise floor above 30 dB (note the upper levels of NSA and SGP with 0 modes). The improved layer selections will now be represented in Figure 4 of the revision, and below in Figure R3.

[Figure]

Fig. R3: Updates to the previous iteration of figure 4, with corrections and adjusted layer selections. As before the caption will read: "Radar moment MDV and SW for three times for each of the three cases: (a) SBRO at (i-ii) 1855 UTC, (iii-iv) 1902 UTC, and (v-vi) 1908 UTC;

(b) NSA at (i-ii) 1521 UTC, (iii-iv) 1531 UTC, and (v-vi) 1537 UTC; (c) SGP at (i-ii) 1245 UTC, (iii-iv) 1248 285 UTC, and (v-vi) 1252 UTC. Purple shading indicates multi-modal layers, yellow is turbulence, and cool green are control layers, unaffected by spectrum-broadening processes. Pink and cyan error bars along each black line is the standard deviation of each moment variable, taken in time over 145-s periods. "

Shown in Figure R3, we retain the original style of human-influenced layer selection (designated turbulent, mode, control) though this is now supplemented by the analysis of the GMM output (which in figures R1 and R2 designated uni-modal and multi-modal heights). The following figure "Fig. R4" is a replacement for former Figure 5 in the preprint. The former heat-map is removed for visual clarity and instead it now contains only the mean and error bar visualizations. This is provided for several subsets of datapoints, in Fig. R3a we show how the contributing cases fill the parameter space when examined via layer classification and via GMM mode identification. In Fig. R3b, we show the same quantities with restricted stdev(MDV) to represent the effect of the use of that parameter to exclude signals caused by turbulence.

As a result, the parameter space of mean moment SW and standard deviation moment MDV have changed. Notably with our increased tolerance for less distinct secondary modes in the parameter identification, there is a greater spread in the possible mean(SW) values, but generally similar values for the uni-modal (control) layer. In the most recent iteration, a multi-modal layer via layer analysis has a mean SW of 0.19 m s$^{-1}$ and a multi-modal layer via the GMM technique has a mean SW of 0.175 m s$^{-1}$. Examining both groups again with likely turbulent layers filtered out (corresponding to a standard deviation of MDV greater than 0.2 m s$^{-1}$) multi-modal layer via layer analysis has a mean SW of 0.19 m s$^{-1}$ and a multi-modal layer via the GMM technique has a mean SW of 0.165 m s$^{-1}$. While this is close to our previously indicated criterion of 0.19 m s$^{-1}$, we plan to re-run the results of Part 2 with an adjusted value of 0.17 m s$^{-1}$ applied. We expect this to somewhat increase the total number of cases and case lengths, and to potentially detect cases that do not meet the previous verification criteria as outlined in both initial submissions.

Examining these added figures, it is clear that the GMM analysis is able to discern more subtle modes that do not always meet our criteria for what we considered as "distinct" with a detectable drop in reflectivity between the modes (see line 263 in the preprint of part 1: a 5 dB or larger drop in reflectivity between the modes). This new analysis has given us this additional reference point for the potential for a user to adjust their SW criterion when interested in detecting more subtle secondary and tertiary modes, and for multi-modal events that this criteria may miss.

We have modified the revised text to explain this revised approach and to make the points discussed above regarding our methods and associated with the revised figures.

[Figure]

Fig. R4: (a) Comparison of uni-modal and multi-modal layers by two methods – uni- and multi-modal parameters by case plotted with error bars, with their aggregate average plotted by stars. The markers at the center of the cross-hairs represent the average $SW$ and $\sigma(MDV)$ for the set of points in each category. The extent of the cross-hairs in each direction represents the standard

deviation $SW$ and $\sigma(MDV)$ of for the set of points in each category. Similarly (b) contains the same results with values restricted to stdev(MDV)<0.1 m s$^{-1}$ (our turbulence threshold).

Unshown in this comment, we also tried k-means clustering as an unbiased method to categorize and classify the regions of the parameter space, but opted for the GMM results as they have a mathematical connection to the modality, though we noticed the clustering method also suggested a distinction between categories at mean(SW) near 0.15 m/s.

Regarding the classification of turbulent layers: if you have a particular method for detecting those in data from vertically pointing radars, we would love incorporate it. For the time being, we are improving our justification for the identification of turbulence through the use of the standard deviation of MDV.

Turbulence, the irregular fluctuations in the motion of air, is observed in all three velocity components and these fluctuations can then lead to an increased value in the standard deviation of MDV, greater than in non-turbulent layers (AMS Glossary). Increases in the variability of MDV will translate to a larger standard deviation of MDV. MDV is essentially a sum of the contributing velocities associated with motions (e.g. uniform wind, shear, turbulence) and the spectrum width quantifies the spread produced by the sum of those processes (Doviak and Zrnic 1993). Turbulence within a cloud or cloud layer will increase the variability of MDV and localized updrafts and downdrafts may be observed, also broadening the total spectrum width (Borque et al. 2016). The effect of turbulence at a short time scale (i.e. the 3.7 s integration time or 145 s temporal average) is to increase SW and cause rapid oscillations in MDV. When the 145 s period is averaged, those rapid oscillations can result in a maxima of standard deviation of MDV. More uniform processes (occurring consistent across a two-minute period or consistent within the cloud) do not affect the standard deviation as strongly.

We retain the manual classification of turbulent layers for the layer analysis, but now also incorporate a more objective turbulence "cut-off" and examine the GMM-detected multi/uni-modal layer parameter space before and after removing turbulent heights from consideration (see Figure R4). This helps to distinguish the features of multi-modal signals from turbulent signals.

Although the reviewer states that, if we run this algorithm, we would "not need to manually go through a lot of cases in order to visually see if your algorithm is valid." We respectfully disagree and believe that validation is essential to building confidence in the algorithm – in other words, we do not feel comfortable fully relying on the GMM detections as "truth" here. In our ongoing revisions to part 2, we will be including a subset that was tested against the GMM mode detection, however with the extreme computational cost of processing the full three year period at the resolution required, it would be prohibitively expensive computationally and in time for the entire verification analysis.

--------- --------- --------- --------- --------- --------- --------- --------- --------- --------- --------- ---------

**Minor Comments**

Minor Comment 1: (line 82) you are saying that in order to identify microphysical processes in the radar you require additional information such as in-situ obs, or other radar obs. However, neither a radar of in-situ will ever be able to observe processes. The only thing we can observe

with a radar is the effect a process has on several aspects of the observed particle distribution. Maybe rephrase that sentence to make it a less strong statement

Good point! This has been rephrased: "However, attempts at identification of potentially active processes requires additional information, including polarimetry, temperature profiles, and/or in situ data to understand if there are favorable conditions or necessary ingredients for processes (e.g., riming or rime splintering)." The intent of the sentence was to indicate, as you stated, that we can only observe the effects of processes occurring and that radar alone is insufficient to make a claim.

Minor Comment 2: (Introduction) I am missing a paragraph about peak detection algorithms, since you are trying map your number of peaks (single or multi-modal) to radar moments, in my opinion it is important to mention and discuss those peak detection algorithms

A paragraph is now added to our revised paper, detailing the Gaussian mixture approach as described above in response to Major Comment 1 and the separate response to Reviewer 1 Major Comment 1. We also cite alternative tools like peako/peaktree given the reference you provided.

**Minor Comment 1: (Table 1) you are specifying that you are only averaging the KASPR data over 1 second. However, in my experience to reduce noise in the spectra it is necessary to average over at least 3 seconds, better close to 4 seconds. Can you comment on the integration time used?**

Apologies for our lack of clarity here --- when we apply the criteria (set forth in Part 1) to the moment data, we are using ~145-s averaged data, in which much of the noise is smoothed. The numbers you cite from Table 1: this is actually the integration time used to obtain a single spectra "time" and come from the team at Stony Brook and pertains to the temporal resolution of the original observations. Presently the spectra are only used at their native temporal integration in plotting (Figure 1 and Figure 2). When it comes to their use, you are quite correct. When we are processing the spectra with the newly added peak detection, the 1-s (and even 3.7-s) data are noisy and require smoothing. This is expanded on in detail in the pending revision, but, ultimately, we now average 11-12 s of spectra to determine the GMM-computed count of modes.

Minor Comment 3: (Figure 2) in the title you are using SBU for stony brook university, however, in the text you refer to it as SBRO, perhaps it would be good to change that to be consistent

Thank you for pointing this out, this title has now changed for consistency to say SBRO.

Minor Comment 4: (Figure 3) the time display is a bit confusing (i.e. 18.8 as time [UTC] perhaps it would be better to go to HH:MM format)

We agree that this is a confusing way to indicate time, this has been changed to HH:MM format in the revised figure.

Minor Comment 5: (Line 244) how do you know that SW>0.2m/s is enhanced?

This sentence has been rewritten for clarity. We are referring to the regions of SW that are greater than the background SW that are visualized after this section, and then later quantified. The sentence now reads: "Given that both the multi-modal spectra and these turbulent layers feature *wider spectra spanning a broad range of velocity bins*, we seek additional information from the integrated moments to help distinguish between these two types of layers."

**Minor Comment 6: (Line 297) you say you are using 499 data points. How many uncorrelated layers of turbulence and multi-modalities are these?**

The count of data points refers to the breakdown in Table 2; displaying the count of data points from the contributing cases as indicated in the highlighted layers visualized in Figure 4. The contributing layers and thus total contributing data points has changed with the updates to our methodologies for obtaining the designated layers from the foundational cases. The times used for each case are separated such that they do not share contributing points, but do share the same clouds and precipitation systems active within the times used. Data from SBRO are independent from the other sites, and in turn NSA and SGP are as well. To address the more manual manner in which layers were selected for closer analysis and parameter values, we also now include that analysis for the entire depths of the cases via GMM (see Figure R3)

**Minor Comment 7: (Line 338) you are saying you are restricting your cases to MDV←2m/s. From your Figure 6 I am taking that you have LDR available correct? Perhaps it would be beneficial to first do a melting layer detection using LDR (LDR strongly increases at the edge of the ML, allowing a detection of the ML), and then afterwards apply your multi-modality detection algorithm to all data above the ML. Then you would also be able to use the algorithm on cases with graupel**

Thanks for this comment, this is a good way to increase the usability of this method. An additional parameter within the function for processing radar moment data has been added such that when LDR is available (as for ARM KAZR radars) that method will be used for moment processing to identify potential multi-modal layers. We will retain the discussion of the use of MDV for this in the case that someone would wish to apply this method without the availability of LDR.

For the purposes of this study, we will adopt the LDR criteria. See Figure R5 for an example of this new threshold. We evaluated some known rain events from the verification study, shown is 17 August 2023, and confirmed a reasonable value of the slope of LDR. Note that because we examine 145 s periods for this study, we determined our LDR slope threshold using the averaged data. The effects of this comment are more prominently clear in the revisions to the manuscript for part 2.

[Figure]

Figure R5: KAZR data from 17 August 2023, in which the MDV rain filter previously was activated at MDV > 2 ms-1, indicated by the dashed black line. Panel (a) shows mean Doppler velocity, panel (b) shows LDR, panel (c) shows the rate of change of LDR at three selected times during the hour.

---

## Author Response (AR2)

egusphere-2025-671: Detection of Multi-Modal Doppler Spectra. Part 1: Establishing Characteristic Signals in Radar Moment Data

**Authors Response – Summary of Changes:**

Figures have had DPI increased and text/label sizes increased. The paper is now using the provided Latex template rather than the Word template to accommodate the table formatting without the insertion of images.